# Spatial Analysis: A Socioeconomic View on the Incidence of the New Coronavirus in Paraná-Brazil

**Elizabeth Giron Cima** [1,*], **Miguel Angel Uribe Opazo** [1,†], **Marcos Roberto Bombacini** [2,†],
**Weimar Freire da Rocha Junior** [1,†] **and Luciana Pagliosa Carvalho Guedes** [1,†,‡]

1   Centro de Ciências Exatas e Tecnológicas (CCET), Western Paraná State University (UNIOESTE),
    Cascavel 85819-110, PR, Brazil
2   Coordenação do Curso de Graduação em Engenharia Eletrônica (COELT), Federal University of
    Technology—Paraná (UTFPR), Curitiba 85902-490, PR, Brazil
*   Correspondence: elizabeth.cima@unioeste.br; Tel.: +55-(45)-32203000
†   These authors contributed equally to this work.
‡   Current address: Rua Universitária, 1619-Bairro Universitário, Cascavel 85819-110, PR, Brazil.

**Abstract:** This paper presents a spatial analysis of the incidence rate of COVID-19 cases in the state of Paraná, Brazil, from June to December 2020, and a study of the incidence rate of COVID-19 cases associated with socioeconomic variables, such as the Gini index, Theil-L index, and municipal human development index (MHDI). The data were provided from the Paraná State Health Department and Paraná Institute for Economic and Social Development. For the study of spatial autocorrelation, the univariate global Moran index (I), local univariate Moran (LISA), global Geary (c), and univariate local Geary ($c_i$) were calculated. For the analysis of the spatial correlation, the global bivariate Moran index ($I_{xy}$), the local multivariate Geary indices ($C_i^M$), and the bivariate Lee index ($L_{xy}$) were calculated. There is significant positive spatial autocorrelation between the incidence rate of COVID-19 cases and correlations between the incidence rate of COVID-19 cases and the Gini index, Theil-L index, and MHDI in the regions under study. The highest risk areas were concentrated in the macro-regions: east and west. Understanding the spatial distribution of COVID-19, combined with economic and social factors, can contribute to greater efficiency in preventive actions and the control of new viral epidemics.

**Keywords:** risk areas; virus spread; correlation indexes; preventive measures; decision making

## 1. Introduction

SARS-CoV-2 (Severe acute respiratory syndrome 2) is a virus from the coronavirus family that causes acute and severe infections in the respiratory tract, and when infecting humans, causes a disease called COVID-19. Because it was a microorganism that, until recently, was not transmitted between humans, it became known, at the beginning of the pandemic, as the "new coronavirus" (Andersen et al., 2020). Initially, the first information about the new coronavirus was related to China, according to reports until December 2019, also called COVID-19, it became known worldwide between January and February 2020, when it was flagged by the World Health Organization (WHO), as a risk of a world order epidemic. In that regard, it was necessary to adopt emergency measures (individually and collectively) in restrictive ways, requiring effort and commitment from the population, which was proposed by Zhu et al. [1].

Quinino et al. [2] report that the spread and unbridled advance occurred quickly and continuously in different countries, which became a challenge for all of them, especially in those where social inequality is more evident.

Recent studies point out that where this inequality is more evident, factors such as basic sanitation, access to education, and health conditions, among others, have an impact on the spread of the new coronavirus, which is more devastating. In this sense, following health

recommendations is an urgent initiative for these scenarios, see Palú et al. [3]. In Brazil, the first cases of the new coronavirus (COVID-19) were confirmed in mid-February 2020 and, from that date onwards, the spread occurred in an alarming way, putting the entire system at risk, whether: economic, political, social, or environmental. The impact that the pandemic causes is evident when analyzing the instability it causes in different segments of the economy, which is proposed by Shang et al. [4].

These scenarios are associated with socioeconomic factors that worsen when considering the economically less favored regions where the offer of services related to health and social well-being is more restricted and limited, as is the case of the most distant neighborhoods, peripheries, and rural areas that are more isolated from development centers; see Arruda et al. [5].

From February 2020, initially slowly, but with exponential growth, transmissions of COVID-19 were detected in municipalities of Paraná, in which the negative impacts affected several segments such as the economy and health, which was overwhelmed, see Lana et al. [6] and Banhos et al. [7]. The regions that were affected had differences in the spread of the disease, in addition, the incidence rates represent a high risk of mortality, which is proposed by Banhos et al. [7]. Initially, the disease was restricted to the east health macro-region, later advancing to the central north, Campos Gerais, and west regions, which corroborates SESA [8]. Along those lines, Pedrosa and Albuquerque [9] argue that it is important to know the geographic space in which the disease spreads, as well as to identify its clusters.

Soltani et al. [10] report that in order to identify groups with greater vulnerability in a pandemic situation, it is important to analyze socioeconomic variables as they are necessary because they have the ability to explain and show ways to develop measures to favor public health.

For the statistical study of spatial patterns and correlations of the areas, the literature presents the univariate Moran index ($I$), the local Moran index ($I_i$), and the bivariate Moran index ($I_{xy}$), respectively. It also presents the univariate global index of Geary (c), local Geary ($c_i$), and local Geary multivariate index ($C_i^M$) for the study of spatial correlation of quantitative variables considering georeferenced data, which is proposed by Cima et al. [11]. Anselin [12] proposes a multivariate extension of the local Geary statistic ($C_i^M$) that measures the proximity of neighboring points in the multidimensional variable space by correlating them with their neighbors in the geographic space. Thus, multivariate local Geary's statistic ($C_i^M$) measures the average distance of an attribute's values and its ability to influence the values of its neighbors under spatial randomness.

In this sense, it becomes important and necessary to understand the different behaviors of the geographic space in which individuals and scenarios are inserted to better understand and analyze possible patterns, see Castro et al. [13].

The objective of this study was to analyze the spatial autocorrelation of the case incidence rate and the spatial correlation between the case incidence rate with the socioeconomic variables, namely: the Gini index, Theil-L index, and municipal human development index (MHDI) in the state of Paraná, Brazil, from June to December 2020.

This is a quantitative descriptive study with a spatial outline that portrays three objectives (health and well-being) as well as eleven objectives (sustainable cities and communities) of the United Nations Program for Sustainable Development (UNDP), whose units of analysis were formed by the three hundred and ninety-nine municipalities that are in the twenty-two regions of health, which are part of the municipalities that make up the state of Paraná.

## 2. Materials and Methods

Moran's global spatial autocorrelation index ($I$) allows evaluation of the global autocorrelation, i.e., it provides a single value for the entire study area, whereas Moran's local index ($I_i$) measures the spatial correlation in each Moran location [14].

Global and spatial autocorrelation were calculated using the univariate global Moran index($I$) and univariate local Moran index ($I_i$), with the global and local Moran index ranging from $-1$ to 1, according to the methodology of Moran [14]. With the univariate global Geary index ($c$) and univariate local Geary ($c_i$), it is assumed that the spatial autocorrelation depends on the distance between two or more observations, its values vary between 0 and 2, and if $c = 0$, it indicates positive spatial autocorrelation; if $c = 1$, it indicates the absence of autocorrelation and if $c > 1$, it indicates negative spatial autocorrelation, see Anselin [14] and Anselin [15], according to Equations (1) and (2), respectively.

Geary's global index ($c$) proposes the evaluation of the global association, assuming that the spatial association depends on the distance between two or more observations. Geary's local association index measures the degree of spatial correlation at each specific location, i.e., provides a value for each Anselin location [15].

Geary's local index ($c_i$) is a spatial association statistic called LISA because it satisfies two criteria: the applicability of each observation to point out statistically significant spatial clusters, and the ability to show that the sum of each region analyzed is proportional to Geary global spatial association index, see Anselin [12], according to Equations (1) and (2), respectively.

$$c = \frac{(n-1)}{2\sum\limits_{i=1}^{n}\sum\limits_{j=1}^{n} w_{ij}} \frac{\sum\limits_{i=1}^{n}\sum\limits_{j=1}^{n} w_{ij}(x_i - x_j)^2}{\sum\limits_{i=1}^{n}(x_i - \bar{x})^2}, \tag{1}$$

where,

$n$: number of spatial units (municipalities);

$x_i$ and $x_j$: X attribute values considered in regions $i$ and $j$;

$\bar{x}$: average value of attribute X in the study region;

$w_{ij}$: element of the normalized neighborhood matrix, corresponding to spatial weights 0 and 1, being 0 for areas $i$ and $j$ that do not border each other and 1 for areas $i$ and $j$ that do border each other, see Anselin [16].

$$c_i = \sum_{j=1}^{n} w_{ij}(x_i - x_j)^2, \tag{2}$$

where,

$(x_i - x_j)^2$: represents the square of the difference between the pairs of values of the attribute under study.

In Equations (1) and (2), to test the significance of global and local spatial autocorrelation, the pseudo-significance hypothesis test $Z(c)$ was used; see Almeida [17] and Cima et al. [11].

Moran's bivariate index ($I_{xy}$) is an index of spatial correlation between two variables ($X$ and $Y$) that are obtained in $n$ spatial units, see Moran [14] and Almeida [17].

In the study of spatial correlation, Almeida [17] reports that, in the bivariate analysis, it is sought to identify whether there are patterns of spatial association between two spatially georeferenced variables. The bivariate Moran index ($I_{xy}$) calculated is an index of spatial correlation between two variables ($X$ and $Y$) that vary from $-1$ to 1 and are obtained in n spatial units. Moran's bivariate index ($I_{xy}$) is presented in Equation (3):

$$I_{xy} = \frac{\sum\limits_{i=1}^{n}\sum\limits_{j=1}^{n} u_i z_j w_{ij}}{S_0 \sqrt{S_u^2 S_z^2}}, \tag{3}$$

where,

$z_j$ and $u_i$: values centered on the average values of the variables under study X and Y, respectively, i.e., $z_j = (x_j - \bar{x})$ and $u_i = (y_i - \bar{y})$;

$w_{ij}$: element of the normalized neighborhood matrix, corresponding to spatial weights 0 and 1, being 0 for areas $i$ and $j$ that do not border each other and 1 for areas $i$ and $j$ that do border each other;

$S_0$: sum of the elements $w_{ij}$ of the symmetric matrix of spatial weights $W$;

$S_z^2$ and $S_u^2$: correspond respectively to the variances of $X$ and $Y$.

Geary's local multivariate index as presented in Anselin [16] is defined as a multivariate statistic that is used to measure the tension between the similarity of variables and the similarity of geographic locations.

Multivariate statistics are based on distances in a higher dimensional space. Multivariate statistics, therefore, provide an additional perspective to measure the tension between the attribute and location similarity, see Anselin [12].

Geary's multivariate local statistic $(C_i^M)$, presented in Equation (4) is defined as the sum of the individual statistics for each of the variables under analysis, with $k$ variables, ordered by $h$, as follows:

$$C_i^M = \sum_{h=1}^{k} \sum_{j=1}^{k} w_{ij} \left( x_{hi} - x_{hj} \right)^2, \tag{4}$$

where,

$k$: total number of variables under study;

$h$: represents each variable at each iteration.

This metric corresponds to a weighted mean of the squared distances in the multidimensional space of attributes between the values observed in a given geographic location $i$ in relation to its geographic neighbors $j \in N_i$ (with $N_i$ as the neighborhood set of $i$), see Anselin [12].

Lee [18] considers that the bivariate analysis seeks to identify the existence of patterns of spatial association between two variables. In this sense, the same author proposed a measure of parametric spatial association to assess bivariate spatial dependence.

Lee's bivariate global index $(L_{xy})$ was also calculated accordingly to Lee [18], who proposed a measure of parametric bivariate spatial association to assess bivariate spatial dependence by integrating Pearson's correlation coefficient and Moran's index I, assuming values from $-1$ to 1, the closer to 1 the greater the positive spatial correlation between the variables, which is proposed by Lee [18] and Lee [19]. Lee's bivariate global index $(L_{xy})$ is presented in Equation (5):

$$L_{xy} = \frac{n}{\sum\limits_{i=1}^{n} \left( \sum\limits_{j=1}^{n} (w_{ij})^2 \right)} \frac{\sum\limits_{i=1}^{n} \left[ \left( \sum\limits_{j=1}^{n} w_{ij}(x_i - \bar{x}) \right) \left( \sum\limits_{j=1}^{n} w_{ij}(y_i - \bar{y}) \right) \right]}{\sqrt{\left( \sum\limits_{i=1}^{n} (x_i - \bar{x})^2 \right)} \sqrt{\left( \sum\limits_{i=1}^{n} (y_i - \bar{y})^2 \right)}}, \tag{5}$$

where,

$x_i$ and $y_j$: values of continuous variables $X$ and $Y$ considered in area $i$ and $j$;

$\bar{x}$, $\bar{y}$: is the average value of variables $X$ and $Y$ in the region under investigation.

The state of Paraná is located in the southern region of Brazil and has approximately 11,597,484 inhabitants and an approximate area of 199,307,922 km$^2$, according to IBGE [20]. The spatial distribution of health services is represented by twenty-two health regions, which are grouped into six macro-regions of care, located in the east, Campos Gerais, central-south, west, northwest, and north, see SESA [8], as shown in Figure 1.

Fiocruz [21] reports that the low testing of the population may be related to social inequality in the regions.

Starling et al. [22] informs that the incidence rate of cases per 100.000 inhabitants in 14 days is defined by calculating the slope angle of the adjustment line, considering the rates of new cases in the last five 14-day period (last 10 weeks).

Alves et al. [23] report that COVID-19 is a highly severe viral disease and represents a public health problem.

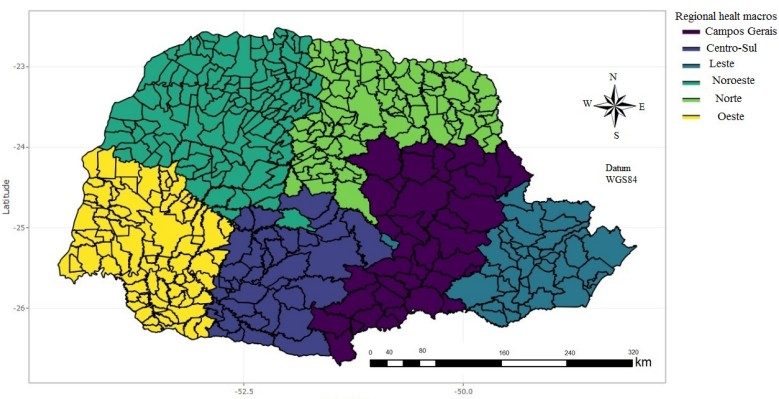

**Figure 1.** Location map of the Paraná Health macro−regions.

As defined by the World Health Organization, it is recommended that tests be carried out in suspected cases, i.e., in more advanced cases as well as in people who have symptoms of respiratory diseases or the flu; WHO [24].

The variable of interest for the study was the incidence rate of cases of the new coronavirus for the month of June to December 2020 whose units of analysis are formed by the three hundred and ninety-nine municipalities that are in the twenty-two regions of health that are part of the municipalities that make up the state of Paraná. The socioeconomic variables associated with the variable of interest were: Gini index, Theil-L index, and municipal human development index (MHDI) see IPARDES [25]. The incidence coefficient of the number of cases was calculated considering the cases that occurred from June 2020 to December 2020, after consulting the Paraná State Health Department (SESA), based on public epidemiological reports, according to Equation (6):

$$Coefficient_{incid} = \frac{Number\ of\ confirmed\ cases\ of\ COVID-19}{Total\ resident\ population\ in\ the\ period\ 2020} \times 100.000\ inhabitants. \tag{6}$$

The Gini index is defined as a measure of inequality, it is used to calculate the inequality of income distribution, see IPEA [26].

The Gini index was obtained from the Paraná Institute of Economic and Social Development (IPARDES), it measures the degree of income concentration in a given group, presents the difference between the income of the poorest classes and the income of the richest classes, and the calculation is obtained according to Equation (7):

$$G = \frac{\sum\limits_{i=1}^{n}\sum\limits_{j=1}^{n}\left|x_i - x_j\right|}{2n^2\bar{x}}, \tag{7}$$

where,

$x_i$: income of municipality $i$;

$x_j$: income of municipality $j$;

$n$: total number of municipalities;

$\bar{x}$: average of the total income of the state of Paraná, divided by the number of municipalities.

The Gini index ranges from 0 to 1, where 0 indicates that all individuals have the same income and 1 only one individual has all the wealth, see IPEA [26].

The Theil index is defined as a measure of inequality based on the uncertainty of the economic distribution, see Theil [27].

Theil proposed two measures of inequality, the Theil-T measure, considers that the weighting factors of inequality within groups are the appropriate income fraction, whereas, in the Theil-L measure, the weighting factors of income inequality within groups are the populations of the groups. For this work, the Theil-L inequality measure was considered, accordingly to IPARDES [25]. It is presented in Equation (8):

$$L = \sum_{i=1}^{n} \frac{1}{N} \log \frac{\mu}{x_i},$$

(8)

where,

$N$: proportion of the population (municipality) corresponding to the $h$-th group;

$n$: group municipality (macro-regional);

$\mu$: average income of the population (municipality) within the group;

$x_i$: income of the $i$–th municipality.

The human development index was developed by the United Nations Development Program (UNDP), it is used to measure the human development of countries which is measured from three dimensions: access to culture and education, quality of life, and standard of living. adequate income. The indicators used to measure this dimension are: literacy rate, life expectancy at birth, and per capita income. Equation (9) shows the formula for the human development index by Municipality which was adapted from the United Nations Development Program, see UNDP [28]

$$MHDI = \frac{MHDI_S + MHDI_E + MHDI_R}{3},$$

(9)

where,

$MHDI_S$: healthy life index;

$MHDI_E$: index of access to education and culture;

$MHDI_R$: adequate income standard of living index.

The human development index by municipality is based on the estimation of economic, political, cultural, and social aspects, it is used to quantify the development of a certain population group, see IPARDES [25].

Moreover, through the Paraná Institute of Economic and Social Development (IPARDES), the municipal human development index was obtained, which is an estimate of each municipality in the year 2010. It can be classified as very low (0 to 0.499), low (0.500 to 0.599), medium (0.600 to 0.699), high (0.700 to 0.799), and very high (0.800 to 1), see Dawalibi et al. [29]. It is considered a long-term measure for the development of the categories: education, evaluation, and income; see IPARDES [25]. The epidemiological idea used here is focused on understanding the development of the transmission of the new coronavirus by the municipality of Paraná and its health macro-regional areas, where we sought to characterize and identify these transmission patterns.

For the construction of thematic maps, georeferenced data by municipalities of Paraná were adopted, in the shape file format in WGS84 geographic coordinates, see IBGE [20]. The R 4.5.0 program [30] and the QGIS software version 3.10 [31], see Anselin [12] were used interactively for the variables: COVID-19 case incidence rate, Gini index, Theil-L index, and municipal human development index (MHDI).

## 3. Applications of Analysis of Data Spatial of the Incidence Rate of COVID-19 Cases

The Moran and Geary indexes: Global and local, were applied for the analysis of univariate spatial autocorrelation and the Moran index and the Lee Global index for the analysis of the bivariate spatial correlation, as well as Geary's local multivariate index. These indices allow the identification of areas with high, medium, and low spatial dependence, as well as identifying whether there is a significant spatial association between the two variables analyzed. Furthermore, they made it possible to identify regions in which there is no statistically significant spatial autocorrelation, An overview of this methodology can be found in the works of Moran [14], Anselin [15], and Anselin [16].

Spatial data analysis techniques were used, namely: the global and local spatial auto-correlation index (LISA) univariate of Moran and Geary, global bivariate spatial correlation of Moran ($I_{xy}$), multivariate local correlation of Geary ($C_i^M$), and Lee's bivariate global correlation ($L_{xy}$). The geolocations of the three hundred and ninety-nine municipalities that make up the state of Paraná were considered. This database has the following data: Geocod-ing of each municipality, the name of the municipalities, and the geographic coordinates (Latitude and Longitude) of the variables studied. For the study of spatial data analy-sis, socioeconomic variables were used, as presented in Section 2 and are represented in Equations (6)–(8), see Dawalibi et al. [29], IPARDES [25], IPEA [26], SESA [8] and Theil [27].

For the study of spatial autocorrelation, the global univariate Moran index ($I$), univari-ate Moran local (LISA), global Geary ($c$), and univariate Geary local ($c_i$) were calculated. For the analysis of the spatial correlation, the global bivariate Moran index ($I_{xy}$), the local multivariate Geary indexes ($C_i^M$), and the bivariate Lee index ($L_{xy}$) were calculated. In ad-dition, the location of each of the six health macro-regions was considered, considering the 399 municipalities that make up the state of Paraná Brazil through the geocode of each municipality and the variables analyzed were: the incidence rate of COVID-19 cases associated with socioeconomic variables, namely: the Gini index, Theil-L index, and human development index by municipality (MHDI).

### 3.1. Results of the Univariate Global Moran Index (I), Local Univariate Moran (LISA), Global Geary (c), and Univariate Local Geary ($c_i$)

Equations (1) and (2) the formula for the global Geary index (c) and local Geary index ($c_i$), are presented in Section 2. The use of these equations was necessary to calculate the autocorrelation, as well as to show their presence or absence between neighboring municipalities and to present the clusters (when they exist), in order to direct possible decision-making and preventive actions, inherent to the reality of each macro-region of health. For this construction, it was necessary to elaborate a spatial proximity matrix that presents the weights for each situation, being neighbor = 1 and not neighbor = 0, called the Queen contiguity criterion, see Anselin [16]. For the analysis of the global and local index by Moran and Geary, the results are shown in Figures 2–4.

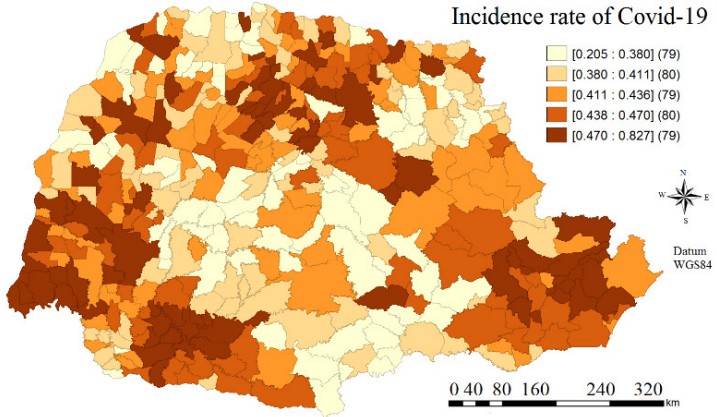

**Figure 2.** Map of the spatial evolution of the incidence rate of COVID-19 cases by the municipality, stratified by quantile intervals, referring to June to December 2020.

From the map of the spatial evolution of the incidence rate of COVID-19 cases by the municipality, stratified by quantile intervals, referring from June to December 2020, in Figure 2, it is observed that the largest class interval of the incidence rate of cases was between [0.470; 0.827] represented in intense dark brown color, is located in the municipalities belonging to the east macro-regional health that comprises the metropolitan region of Curitiba; north, northwest, and west, which corroborates Barboza et al. [32].

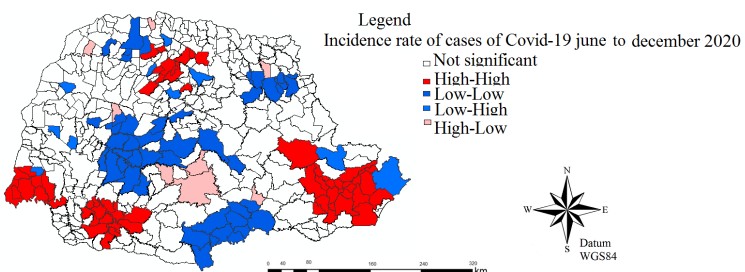

**Figure 3.** Clustering map of the incidence rate of confirmed cases of the new coronavirus by municipalities in Paraná from June 2020 to December 2020, using the Moran local index (LISA).

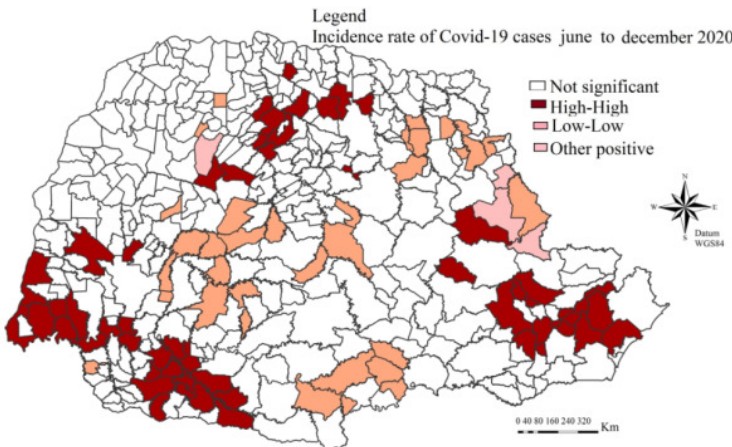

**Figure 4.** Clustering map of the incidence rate of confirmed cases of the new coronavirus by municipalities in Paraná from June 2020 to December 2020, through the analysis of the local Geary index ($c_i$).

As a result, it is observed that there was a positive and significant global spatial autocorrelation of Moran $I$ = 0.325 ($p$-value = 0.001), for the incidence rate of the number of COVID-19 cases in the municipalities and macro-regions of health in the state of Paraná. This spatial profile suggests that in Paraná there are municipalities with high and or low case incidence rates, surrounded by municipalities that present similar characteristics. By calculating the incidence rate of COVID-19 cases using the Moran $I_i$ local autocorrelation index (LISA), the result shows the municipalities belonging to the macro-regional health: east, north central, and west presented clusters of municipalities with high rates of incidence, surrounded by regions also with a high incidence rate (red color in Figure 3).

Yao et al. [33] analyzed the spatial dependence of the spread of COVID-19 in China using the Moran global and local index (LISA) and concluded that there was a significantly positive global spatial autocorrelation of COVID-19 in forty-nine Chinese cities, which were illustrated using maps of local spatial association indicators (LISA). This behavior demonstrates that in Paraná there are municipalities with high and or low case incidence rates, surrounded by municipalities with similar characteristics.

The result showed that the value of Geary's global spatial autocorrelation index for the years studied was c = 0.635 ($p$-value = 0.0001), a value close to zero, indicating that the neighboring regions $i$ and $j$, on average, present the square of the difference as a function of distances, small (values close to zero), showing the presence of positive and significant spatial autocorrelation, see Anselin [15]. From an epidemiological and economic point of view, this information is important and necessary because it allows showing the reality of the panorama of these locations, thus highlighting the social and economic challenges of regions with significant incidence rates, allowing health managers to better understand the spread of the disease in these locations, for possible decision-making.

Recent works, such as Cavalcante and Abreu [34], described the spatial distribution of the first confirmed cases of COVID-19 in the state of Rio de Janeiro through the analysis

of spatial autocorrelation, concluding that there is a high risk of infection by COVID-19 in different regions of the city of Rio de Janeiro and found a high rate of cases of the disease. As seen in Figure 4, in the clustering map for the case incidence rate, the clusters of municipalities with the highest incidence coefficients (High-High), through the local Geary index ($c_i$) represented in dark brown color, with neighbors also with high incidence rates of cases are concentrated in the Campos Gerais, the metropolitan region of Curitiba, southwest, north central, and west regions, while those with low incidence coefficients surrounded by neighbors with low incidence rates, represented in the nude color, are found in the health macro-regions: southeast, south, and central north, as seen in Figure 4 in the clustering map for the case incidence rate.

It is also observed the presence of municipalities with Low-Low spatial autocorrelation, i.e., municipalities with a low incidence rate of the number of cases surrounded by municipalities also with a low incidence rate. This spatial pattern is mainly observed in the macro-regional region of Campos Gerais (in nude tone).

The clustering map (Figure 4) suggests that the region with the highest risk of occurrence for COVID-19 was the macro-regional east (metropolitan region of Curitiba), followed by the southwest, west, and part of the north-central (dark brown tone). The analysis of the local spatial autocorrelation of the case incidence rate made it possible to identify clusters of risk for the occurrence of COVID-19 in certain regions of the studied area, in the analyzed period.

*3.2. Results of the Global Bivariate Moran Index for the Incidence Rate of COVID-19 Cases and the Gini Index, Incidence Rate and Theil-L Index, and Incidence Rate and Human Development Index by the Municipality That Brings Together the Six Health Macro-Regions of Paraná*

The bivariate approach of the global Moran index was used, which is presented in Section 2. As a result of the analysis of Moran's bivariate spatial correlation, the incidence rate of COVID-19 cases in the state of Paraná, Brazil, from June to December 2020 was considered a variable of interest, justifying the intention of this study by the fact that to verify if there is a spatial association between the incidence rate and the socioeconomic indices in the three hundred and ninety municipalities that make up the six macro-regional health regions of Paraná, Brazil, namely: incidence rate of COVID-19 cases and the Gini index, rate of incidence of the Theil-L index, and incidence rate and MHDI. The analysis was performed with the data that were presented in Section 2, and the results are presented in Table 1.

Through Table 1, it was possible to verify that there was a significant negative spatial Moran bivariate correlation ($I_{xy}$) of the incidence rate with the Gini index and incidence rate with the Theil-L index.

Regarding social inequality, the negative spatial correlation between the incidence rate of COVID-19 cases and the Gini index and the incidence rate of cases and the Theil-L index, the result is justified by the fact that the Poor income distribution in a given region can trigger an increase in the incidence rate of cases, in which the condition of social exclusion can be accompanied by low control of health prevention, such as the use of masks and alcohol gel.

This table allowed, indirectly, to infer an influence of this correlation, suggesting that in the regions studied there are important differences between the municipalities and regions of macro-regional health. The disparities identified are justified by several factors, such as the initial date of disease progression in the municipalities and macro-regions of health and the intensity of disease advancement in the different municipalities, which corroborates SESA [8].

**Table 1.** Bivariate Moran global spatial correlation ($I_{xy}$) of COVID-19 Case incidence rate and the Gini index; incidence rate and Theil-L index and incidence rate and municipal human development index (MHDI) in the period from June to December 2020 in the municipalities of Paraná.

| Variables | Moran Bivariate Index Value ($I_{xy}$) | *p*-Value |
|---|---|---|
| Incidence rate and the Gini index | −0.098 * | 0.001 |
| Incidence rate and Theil-L index | −0.100 * | 0.001 |
| Incidence rate and municipal human development index | 0.193 * | 0.0001 |

*: Statistically significant at the 5 % probability level.

There was a significant positive spatial correlation between the incidence rate and the municipal human development index (MHDI), the result suggests that there are regions with a high incidence rate of cases that tend to be surrounded by neighboring regions with a human development index by high municipalities, as well as regions with a low incidence rate of the number of cases surrounded by neighbors with a low MHDI.

By analyzing the spatial correlation between the incidence rate of cases and the municipal human development index (MHDI), it appears that the behavioral pattern of transmission that took place in Paraná is suggestive of being a result of the spread of the disease in cities whose infrastructure and population growth was not compatible with urban development, as supported by the results of the correlation analysis.

The panorama that was found seems to follow the global trend that the transmission of the virus on a large scale occurs in economically disadvantaged populations and compromises more those places with greater social inequalities and low human development index per municipality Quinino et al. [2].

Therefore, when considering the spatial correlation analysis, it was observed that socioeconomic factors such as differences in the concentration and distribution of income, associated with different conditions of human development in the municipalities (Gini index and municipal human development index) were also related to the incidence rate of COVID-19.

*3.3. Results of the Multivariate Geary Local Index ($C_i^M$) on the Incidence Rate of COVID-19 Cases with the Socioeconomic Variables in the Municipalities that Bring Together the Six Health Macro-Regions of Paraná*

For comparative purposes, the multivariate Geary local index approach was used, which is presented in Section 2. The results are presented using the aforementioned index in the three hundred and ninety-nine municipalities that make up the six macro-regional health regions of Paraná. The spatial association between the incidence rate of COVID-19 cases and the Gini index, the incidence rate, and Theil-L index, and the incidence rate and MHDI were analyzed. The results are shown in Figures 5–7.

Analyzing Figure 5, it is possible to see that one hundred and forty-five municipalities presented positive clusters for the incidence rate of COVID-19 cases and the Gini index, through the multivariate Geary local index ($C_i^M$), thus reflecting that the result is suggestive of similarity between these two analyzed variables, in accordance with Anselin [16], noting that the incidence rate of the number of cases is significant in relation to the concentration of income of a particular group of individuals and which points out the difference between the incomes of the poorest populations in relation to the richest populations (Gini index).

The result suggests the presence of similar characteristics between the incidence rate of COVID-19 cases and the Gini index in the analyzed region.

The result shows that this similarity was greater in the central-south, center east (Campos Gerais health macro-region), and west, in the red hue.

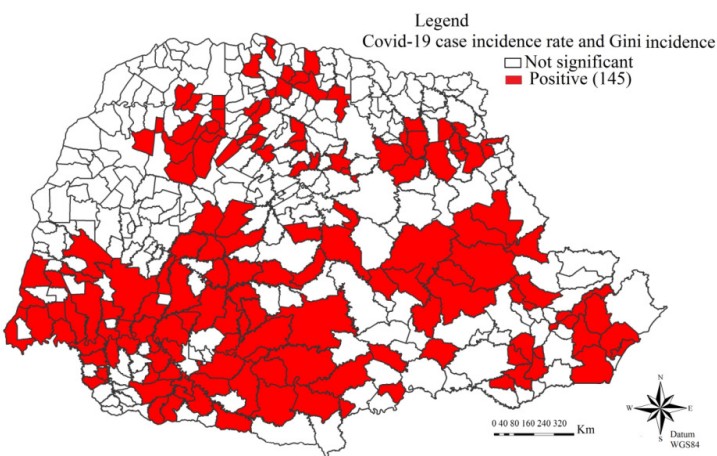

**Figure 5.** Map of the spatial correlation between the incidence rate of the number of confirmed cases for COVID-19 and the Gini index using the multivariate Geary local index $(C_i^M)$, in Paraná.

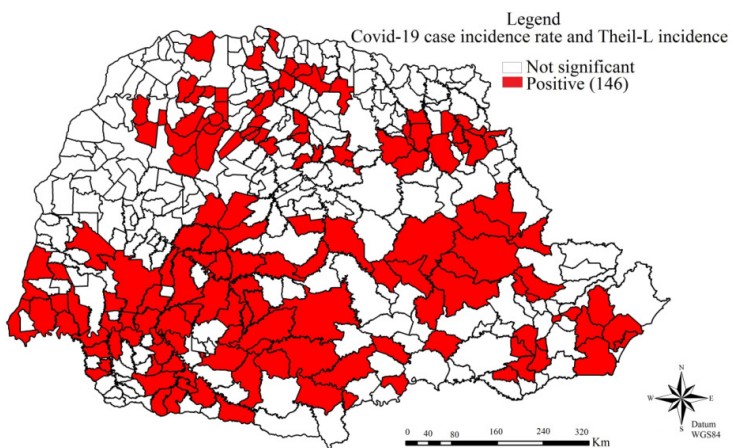

**Figure 6.** Map of the spatial correlation between the incidence rate of the number of confirmed cases for COVID-19 from June to December 2020 and the Theil-L index using the multivariate Geary local index $(C_i^M)$, in Paraná.

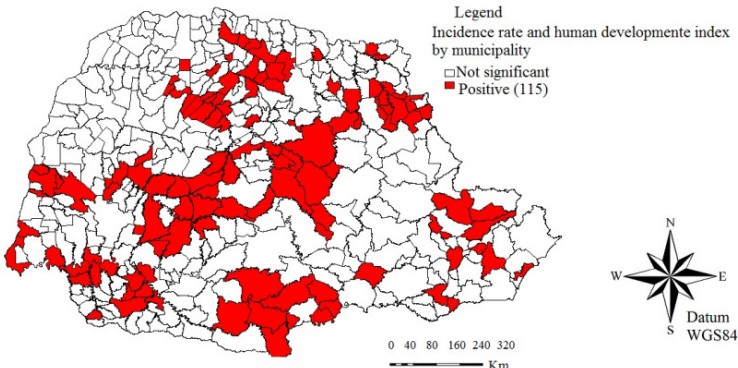

**Figure 7.** Map of the spatial correlation between the incidence rate of the number of confirmed cases for COVID-19 and the MHDI for the months of June 2020 to December 2020 using the local Geary multivariate index $(C_i^M)$, in Paraná.

Oxoli et al. [35] investigated the multivariate Geary local index $(C_i^M)$, in the original data to verify the general behavior of the multivariate spatial association proposed through a mapping procedure with spatial association properties and concluded that the method was efficient in the analysis of spatial data.

Figure 6 shows the spatial analysis of the cluster map using the local multivariate Geary index $(C_i^M)$, the incidence rate, and the Theil-L index; the locations of the observations with the greatest presence of clustering patterns were the Campos Gerais region, part of the central-south, southwest, and west. The result suggests that there is a similarity between the incidence rate of the case numbers for June and December 2020 and the Theil-L index, with the largest clusters being found in the central-south region.

The result showed that in the macro-regional, the largest clusters occurred, evidencing the presence of regions in central-south with high or low case incidence rates and the Theil-L index, surrounded by municipalities with similar characteristics (red color). It is visible through the map that the incidence rate of COVID-19 cases is associated with the distribution of the population's income, places, where income variability is greater, are suggestive of greater social inequality among the population, which corroborates Dell-Olmo et al. [36].

In Figures 5 and 6, it is visible that the characterization of the regions and municipalities of Paraná showed that the municipalities and macro-regions of health showed similarity between the Gini index and the Theil-L index associated with the incidence rate of confirmed cases of COVID-19.

By analyzing Figure 7, the result shows the spatial distribution of the correlation between the incidence rate of the number of confirmed cases of COVID-19 and the municipal human development index through the multivariate Geary local index $(C_i^M)$, in Paraná. There was a positive spatial association between the incidence rate and the MHDI in one hundred and fifteen municipalities in Paraná with cluster formation in the macro-regional health: central-south, Campos Gerais, west, north-central, and north in Figure 7 (red hue), which corroborates Maciel et al. [37].

The result highlights the importance of spatial analysis of epidemiological and socioeconomic data, as it showed the index's ability to present the spatial behavior of the data, see Anselin [15], Freitas et al. [38].

### 3.4. Results of the Global Bivariate Lee's Index for the Incidence Rate of COVID-19 Cases and the Gini Index; Incidence Rate and Theil-L Index and Incidence Rate and Human Development Index by the Municipality That Brings Together the Six Health Macro-Regions of Paraná

The bivariate approach of the global Lee index was used, which is presented in Section 2. The bivariate analysis seeks to identify whether there are patterns of spatial association between two variables under analysis, this proposal was studied by Lee [19] who proposed a measure of parametric bivariate spatial association to assess the bivariate spatial dependence. This index was discussed to verify if there is a spatial association between the incidence rate of COVID-19 and socioeconomic indicators, the result is seen in Table 2.

**Table 2.** Lee's bivariate global spatial correlation $(L_{xy})$ the incidence rate of COVID-19 cases and the Gini index; incidence rate and Theil-L index and incidence rate and municipal human development index (MHDI) in the period from June to December 2020 in the municipalities of Paraná.

| Variables | Lee's Bivariate Index Value $(L_{xy})$ | *p*-Value |
|---|---|---|
| Incidence rate and the Gini index | −0.063 * | 0.010 |
| Incidence rate and Theil-L index | −0.069 * | 0.010 |
| Incidence rate and municipal human development index | 0.224 * | 0.0001 |

*: Statistically significant at the 5% probability level.

Table 2 presents the results from the Lee statistic $(L_{xy})$, as a result, it was possible to observe that there was a significant negative and positive spatial correlation between the variables analyzed in the studied region and that the highest correlation was for the

incidence rate and the human development index per municipality (0.224), thus showing that the human development index per municipality influences the incidence rate of the number of cases.

The result also suggests that there is a relationship between the incidence rate and the municipal human development index and the macro-regional health of Paraná.

Recent studies, such as by Covre et al. [39], reported the importance of spatial data analysis applied in the health area as well as its contribution to scientific research.

The analysis of spatial data through autocorrelation and spatial correlation showed the regions that presented similar characteristics with each other in relation to the incidence rate of COVID-19 cases and the socioeconomic variables analyzed here, suggesting that the adoption of protection and safety measures (regarding health services) is necessary for the smooth running of the health system, which corroborates with finding by Parker et al. [40] and the UNDP [28].

## 4. Discussion

Figure 2 signaled the spatial behavior of the incidence rate of COVID-19 cases by the municipality that make up the six health macro-regions of the state of Paraná, the result made it clear that the highest class range of the case incidence rate was located in the municipalities belonging to the east–north, northwest, and west macro-regional health.

By the univariate Moran and Geary global spatial autocorrelation index, the presence of positive spatial autocorrelation of the incidence rate of the number of COVID-19 cases in the six health macro-regions that make up the state of Paraná, Brazil, was observed, which suggests the presence of regions with high and or low incidence rates of COVID-19 cases surrounded by regions with similar characteristics.

It can be seen that, through the local Moran and Geary index (LISA), $(c_i)$, respectively, this cluster formation profile was observed in the macro-regional health: east, north, and west (Figures 3 and 4). Table 1 shows the presence of Moran's bivariate negative spatial correlation of the incidence rate with the Gini index and of the incidence rate with the Theil-L index.

The results found in Tables 1 and 2, although calculated by different methods of spatial correlation, showed similar values, which highlights the importance of each method used here.

## 5. Conclusions

There was significant positive spatial autocorrelation of the incidence rate of the COVID-19 case numbers across the regions under study.

To the spatial correlation between the incidence rate and socioeconomic index, the analysis treatment showed the presence of a significant negative spatial correlation between the incidence rate and the Gini index and incidence rate and Theil-L index and its effects on neighboring regions which suggests that inequality and social exclusion factors can contribute to the increase in the incidence rate of the number of COVID-19 cases, the same understanding can be applied to the effects of the results obtained for the positive spatial association of the incidence rate and the human development index by municipality (MHDI).

The results found for Paraná presented in this work point out that there are important differences in the distribution of the incidence rate of confirmed cases of COVID-19, among the municipalities and regions of the macro-regional health, which can be justified by factors such as: the start date of the contagion of COVID-19 in the different municipalities associated with their economic infrastructure.

To the spatial correlation between the incidence rate and socioeconomic indices, the analysis treatment showed the presence of a significant negative spatial correlation between the incidence rate and the Gini index and incidence rate and Theil-L index and its effects on neighboring regions, which suggests that inequality and social exclusion factors can contribute to the increase in the incidence rate of the number of COVID-19 cases, the same understanding can be applied to the effects of the results obtained for the positive

spatial association of the incidence rate and the human development index by Municipality (MHDI).

The highest incidence prevalence rates of confirmed cases of COVID-19 in Paraná were observed in the municipalities belonging to the macro-regions of health: east, southwest, and west.

**Author Contributions:** E.G.C., Conceptualization, methodology, editing, software. M.A.U.O., conceptualization, writing—reviewing, supervision. M.R.B., conceptualization, methodology, editing, software. W.F.d.R.J., conceptualization, writing—reviewing, supervision. L.P.C.G., conceptualization, writing—reviewing, supervision. All authors have read and agreed to the published version of the manuscript.

**Funding:** This research was funded by the Coordination for the Improvement of Higher Education Personnel—Brazil (CAPES), the Financing Code 001 and the National Council for Scientific and Technological Development (CNPq).

**Acknowledgments:** The authors are grateful for the financial support given by the Coordination for the Improvement of Higher Education Personnel-Brazil (CAPES), the Financing Code 001, the National Council for Scientific and Technological Development (CNPq), the Graduate Program in Regional Development and Agribusiness at Unioeste–Paraná-Brasil (PGDRA), the Graduate Program in Agricultural Engineering (PGEAGRI), and the Laboratory of Spatial Statistics (LEE/UNIOESTE) at the State University of Western Paraná, Brazil.

**Conflicts of Interest:** The authors declare no conflict of interest.

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
