# Peer review of "Spatial Analysis: A Socioeconomic View on the Incidence of the New Coronavirus in Paraná-Brazil"

_stats, doi:10.3390/stats5040061_

Round 1
Reviewer 1 Report
Abstract:
"This paper presents the spatial analysis of the incidence rate of Covid-19 cases in the state of Paraná-Brazil from June to December 2020, and the study of the incidence rate of Covid-19 cases associated with socioeconomic variables such as Gini index, Theil-L index, and Municipal Human Development Index (MHDI). "
Should be "...a spatial analysis...". The aim here is not compleatly in line with the one in the introduction.
1. Introduction
"The objective of this study was to analyze the spatial evolution of the incidence rate of cases of Covid-19 in the state of Paraná-Brazil, considering the period from June to December 2020."
To general and misses the gini, Theil, MHDI part. It is hard to evaluate if you do the things you are set out to do with such general aim.
You are de facto doing two things:
1: calculating spatial autocorrelation of cases.
2: calculate the correlation between covid cases and the spatial lag of socioeconomic variables.
This could be stated in the aim if this is what you intended to do. The first is rater trivial, it would be expected that there is a spatial autocorrelation when studying a infectious disease. The second thing is a little awkward. It would be more interesting to directly study the correlation between covid cases and socioeconomic situation in the same location. The correlation between a variable and another variables spatial lag is rather unintuitive and hard to interpret. If you want to keep this approach you should put some effort in motivating it and alert the reader that it might be hard to interpret.
2. Materials and Methods
"The variable of interest for the study was the incidence rate of cases of the new coronavirus for the month of June and December 2020. "
How is a case defined? severity? Are all cases tested? Any problems with unequal testing. Maybe death rates are more accurate? June to December 2020
3.2. Results of the global bivariate Moran index
"The result suggests that there are regions with low rates of incidence of the number of Covid-19 cases, tend to be circumcised by regions with a high Gini index and Theil-L index, while regions with a high incidence rate are neighbors of regions with a low Gini index and Theil-L"
The approach leads to conclusions as above leaving the reader wondering; what about the Gini in the same region which of course is more important.
Unable to understand:
" The result presented is suggestive that there is a difference between social classes in which in these regions there are municipalities with high or low incidence rates of the number of Covid-19 cases and Gini index, surrounded by municipalities withsimilar characteristics."
3.4. Results of the global bivariate Lee’s index
To discussion?:
"Similar results were found in Covre et al. [34], who analyzed the spatial correlation between the numbers of confirmed cases of Covid-19 in relation to the number of beds in intensive care units in the municipalities of Paraná and concluded that there is a relationship between confirmed cases of Covid-19 and the distribution of beds and identified the priority areas of attention in the state, related to the dissemination and control of the disease."
I'm unable to see how these conclusions can be drawn from the present work:
"The results found in the present work showed that in a situation of a worldwide pandemic, the protection systems of a nation require initiative to control its dissemination, including political, social, and economic changes, revealing that measures and actions of prevention and sanitary control are present as a challenge for the health sector and the population in general, which corroborates Parker et al."
4. Discussion
Hard to understand especially "its ability to influence neighboring regions". Whats ability to influence what?
" By the global univariate Moran and Geary index, the presence of positive spatial autocorrelation for the incidence rate of the number of cases of Covid-19 in the macro-regional health of the state of Paraná was evident, and its ability to influence neighboring regions. "
Hard to understand language and reasoning:
"A similarity was observed between the incidence rate of cases and socioeconomic indices, making this method interesting in the analysis performed. Also in Table 1, the result suggested a significant positive spatial correlation between the incidence rate and the human development index per municipality, showing regions with a high incidence rate of cases that tend to be surrounded by neighboring regions with a human development index with similar characteristics. This spatial association presented in Table 1 suggests that the transmission pattern of Covid-19 in the state of Paraná is indicative of municipalities with infrastructure with low urban development. From Table 2, the result showed the presence of significant negative and positive spatial correlation between the variables analyzed. The result also suggests that there is a relationship between the incidence rate and the socio-economic indices analyzed here, which were studied in the macro-regional health of Paraná."
5. Conclusions
Here the effect of date is raised. How is this handled in the analysis? How can it affect the analysis results?
Your conclusions are a bit general considering that you are looking at the lag effect of socio-economic variables from the surrounding areas. State that explicitly.
"The disparities identified are justified by several factors, such as the date of the beginning of the contagion in the municipalities and macro-regions and socioeconomic characteristics, evidenced by the Gini index, Theil-L index, and the Municipal Human Development Index (MHDI)"

Author Response
Comments and Suggestions for Authors
Abstract:
"This paper presents the spatial analysis of the incidence rate of Covid-19 cases in the state of Paraná-Brazil from June to December 2020, and the study of the incidence rate of Covid-19 cases associated with socioeconomic variables such as Gini index, Theil-L index, and Municipal Human Development Index (MHDI). "
Should be "...a spatial analysis...". The aim here is not compleatly in line with the one in the introduction.
Answer: The Abstract was aligned with the purpose of the introduction, as suggested by reviewer 1. Please see the adjustments made below:
Abstract: This paper presents the (replaced by a) spatial analysis of the incidence rate of Covid-19 cases in the state of Paraná-Brazil from June to December 2020, and the (replaced by a) study of the incidence rate of Covid-19 cases associated with socioeconomic variables such as Gini index, Theil-L index, and Municipal Human Development Index (MHDI).
- Introduction
"The objective of this study was to analyze the spatial evolution of the incidence rate of cases of Covid-19 in the state of Paraná-Brazil, considering the period from June to December 2020."
To general and misses the gini, Theil, MHDI part. It is hard to evaluate if you do the things you are set out to do with such general aim.
You are de facto doing two things:
1: calculating spatial autocorrelation of cases.
2: calculate the correlation between covid cases and the spatial lag of socioeconomic variables.
This could be stated in the aim if this is what you intended to do. The first is rater trivial, it would be expected that there is a spatial autocorrelation when studying a infectious disease. The second thing is a little awkward. It would be more interesting to directly study the correlation between covid cases and socioeconomic situation in the same location. The correlation between a variable and another variables spatial lag is rather unintuitive and hard to interpret. If you want to keep this approach you should put some effort in motivating it and alert the reader that it might be hard to interpret.
Answer: Reviewer 1's suggestion was followed and the objective of the study was rewritten, in order to make it as explained as possible. Please check the adjustment made below.
The objective of this study was to analyze the spatial evolution of the incidence rate of cases of Covid-19 in the state of Paraná-Brazil, considering the period from June to De- cember 2020. (replaced by: The objective of this study was to analyze the spatial autocorrelation of the case incidence rate and the spatial correlation between the case incidence rate with the socioeconomic variables, namely: Gini index, Theil-L index and Municipal Human Development Index (MHDI) in the state of Paraná-Brazil from June to December 2020.)
- Materials and Methods
"The variable of interest for the study was the incidence rate of cases of the new coronavirus for the month of June and December 2020.”
How is a case defined? severity? Are all cases tested? Any problems with unequal testing. Maybe death rates are more accurate? June to December 2020
Answer: As suggested by reviewer 1:
The incidence rate of the number of cases was defined; Severity has been defined; Test cases were defined; Problems with uneven tests have been defined. Fixed: June to December 2020. Please check the adjustments made below:
Starling et al. [21] informs that the incidence rate of cases per 100.000 inhabitants in 14 days is defined by calculating the slope angle of the adjustment line, considering the rates of new cases in the last five 14-day period (last 10 weeks).
Alves et al. [23] report that Covid-19 is a highly severe viral disease and representes a public health problem.
As defined by the World Health Organization, it is recommended that tests be carried out in suspected case and not only in more advanced cases, it is suggested to carry out testin in people who have symotoms of respiratory diseases ou flu WHO [25].
Fiocruz reports that the low testing of the population may be related to social inequality in the regions FIOCRUZ [28].
3.2. Results of the global bivariate Moran index
"The result suggests that there are regions with low rates of incidence of the number of Covid-19 cases, tend to be circumcised by regions with a high Gini index and Theil-L index, while regions with a high incidence rate are neighbors of regions with a low Gini index and Theil-L"
The approach leads to conclusions as above leaving the reader wondering; what about the Gini in the same region which of course is more important.
Unable to understand:
Response: The paragraph has been adjusted and improved as suggested by reviewer 1. Please see below:
The result suggests that there are regions with low rates of incidence of the number of Covid-19 cases, tend to be circumcised by regions with a high Gini index and Theil-L index, while regions with a high incidence rate are neighbors of regions with a low Gini index and Theil-L. (As suggested by the reviewer, this paragraph has been improved as follows: Regarding social inequality, the negative spatial correlation between the incidence rate of Covid-19 cases and the Gini index and the incidence rate of cases and the Theil-L index, the result is justified by the fact that the Poor income distribution in a given region can trigger an increase in the incidence rate of cases, in which the condition of social exclusion can be accompanied by low control of health prevention, such as the use of masks and alcohol gel.)
" The result presented is suggestive that there is a difference between social classes in which in these regions there are municipalities with high or low incidence rates of the number of Covid-19 cases and Gini index, surrounded by municipalities withsimilar characteristics."
Response: The paragraph has been rewritten and improved as suggested by Reviewer 1. Please see the adjustment below:
The result presented is suggestive that there is a difference between social classes in which in these regions there are municipalities with high or low incidence rates of the number of Covid-19 cases and Gini index, surrounded by municipalities with similar characteristics. (As suggested by the reviewer, this paragraph has been improved as follows):
The result suggests the presence of similar characteristics between the incidence rate of Covid-19 cases and the Gini index in the analyzed region.)
3.4. Results of the global bivariate Lee’s index
To discussion?:
"Similar results were found in Covre et al. [34], who analyzed the spatial correlation between the numbers of confirmed cases of Covid-19 in relation to the number of beds in intensive care units in the municipalities of Paraná and concluded that there is a relationship between confirmed cases of Covid-19 and the distribution of beds and identified the priority areas of attention in the state, related to the dissemination and control of the disease."
Response: The discussion has been revised, reduced, rewritten and improved as suggested by
reviewer 1.
Similar results were found in Covre et al. [34], who analyzed the spatial correlation between the numbers of confirmed cases of Covid-19 in relation to the number of beds in intensive care units in the municipalities of Paraná and concluded that there is a relationship between confirmed cases of Covid-19 and the distribution of beds and identified the priority areas of attention in the state, related to the dissemination and control of the disease.
(As suggested by the reviewer, this paragraph has been improved as follows):
Recent studies such as Covre et al. [39] reported the importance of spatial data analysis applied in the health area as well as its contribution to scientific research).
I'm unable to see how these conclusions can be drawn from the present work:
"The results found in the present work showed that in a situation of a worldwide pandemic, the protection systems of a nation require initiative to control its dissemination, including political, social, and economic changes, revealing that measures and actions of prevention and sanitary control are present as a challenge for the health sector and the population in general, which corroborates Parker et al."
Response: The paragraph was revised and improved, as asked by reviewer 1.
The results found in the present work showed that in a situation of a worldwide pandemic, the protection systems of a nation require initiative to control its dissemination, including political, social, and economic changes, revealing that measures and actions of prevention and sanitary control are present as a challenge for the health sector and the population in general, which corroborates Parker et al. [36] and Pnud [37].
(As suggested by the reviewer, this paragraph has been improved as follows):
The analysis of spatial data through autocorrelation and spatial correlation showed the regions that presented similar characteristics with each other in relation to the incidence rate of Covid-19 cases and the socioeconomic variables analyzed here and suggests that the adoption of protection and safety measures health services are necessary for the smooth running of the health system, which corroborates Parker et al. [40] and UNDP [27].)
- Discussion
Hard to understand especially "its ability to influence neighboring regions". Whats ability to influence what?
" By the global univariate Moran and Geary index, the presence of positive spatial autocorrelation for the incidence rate of the number of cases of Covid-19 in the macro-regional health of the state of Paraná was evident, and its ability to influence neighboring regions. "
Response: The paragraph has been revised and improved. Please see below:
By the univariate Moran and Geary global spatial autocorrelation index, the presence of positive spatial autocorrelation of the incidence rate of the number of Covid-19 cases in the six health macro-regions that make up the state of Paraná-Brazil was observed, which suggests the presence of regions with high and or low incidence rates of Covid-19 cases surrounded by regions with similar characteristics.
Hard to understand language and reasoning:
"A similarity was observed between the incidence rate of cases and socioeconomic indices, making this method interesting in the analysis performed. Also in Table 1, the result suggested a significant positive spatial correlation between the incidence rate and the human development index per municipality, showing regions with a high incidence rate of cases that tend to be surrounded by neighboring regions with a human development index with similar characteristics. This spatial association presented in Table 1 suggests that the transmission pattern of Covid-19 in the state of Paraná is indicative of municipalities with infrastructure with low urban development. From Table 2, the result showed the presence of significant negative and positive spatial correlation between the variables analyzed. The result also suggests that there is a relationship between the incidence rate and the socio-economic indices analyzed here, which were studied in the macro-regional health of Paraná."
Response: The paragraph language and reasoning have been rewritten and improved as suggested by reviewer 1. Please see below:
(As suggested by the reviewer, this paragraph has been improved as follows): The results found in Table 1 and Table 2, although calculated by different methods of spatial correlation, showed similar values, which highlights the importance of each method used here.
- Conclusions
Here the effect of date is raised. How is this handled in the analysis? How can it affect the analysis results?
Your conclusions are a bit general considering that you are looking at the lag effect of socio-economic variables from the surrounding areas. State that explicitly.
"The disparities identified are justified by several factors, such as the date of the beginning of the contagion in the municipalities and macro-regions and socioeconomic characteristics, evidenced by the Gini index, Theil-L index, and the Municipal Human Development Index (MHDI)".
Response: Reviewer's suggestion 1 was accepted. The paragraph has been rewritten and improved considering a more detailed description of the analysis results.
The disparities identified are justified by several factors, such as the date of the beginning of the contagion in the municipalities and macro-regions and socioeconomic characteristics, evidenced by the Gini index, Theil-L index, and the Municipal Human Development Index (MHDI).
(As suggested by the reviewer, this paragraph has been improved as follows):
To the spatial correlation between the incidence rate and socioeconomic index, the analysis treatment showed the presence of a signficant negative spatial correlation between the incidence rate and Gini index and incidence rate and Theil-L index and its effects on neighboring regions which suggests that the inequality and social exclusion factor can contribute to the increase in the incidence rate of the number of Covid-19 cases, the same understanding can be applied to the effects of the results obtained for the positive spatial association of the incidence rate and the human development index by municipality (MHDI).
He results found for Paraná presented in this work point out that there are important diferences in the distribution of the incidence rate of confirmed cases of Covid-19, among the municipalities and regions of the macro-regional health. which can be justified by factors such as: start date of the contagion of Covid-19 in the different municipalities associated with their economic infrastructure.
Response to Reviewer 1 Comments
Point 1: I found that sections 2 & 3 should be re‐organized and be shortened. It may be easier for the readers if the authors define properly the mixture of regression model and the class‐ membership equation first before moving to the computation of the GINI and of the Polarization of subgroups. Sections 2.1 and 2.2 are too long and can be significantly reduced. In section 2.1 the authors assume the condition uk > uj, but this does not appear anywhere else in the calculation of the mixture of regression model. After equation (10) all the other equations are not numbered.
Response 1: Please provide your response for Point 1. (in red)
Response 1: Sessions 2 and 3 have been reorganized, rewritten and shortened. The definitions of each autocorrelation and spatial correlation index worked were added. Please check the adjustments made below
Moran’s global spatial autocorrelation index allows evaluating the global autocorrelation, ie it provides a single value for the entirer study área, whereas Moran´s local index measures the spatial correlation in each location Moran [14]
Geary’s global index ( ) propuses the evaluation of the global association, assuming that the spatial association depends on the distance between two or more observation. Geary’s local association index measures the degree of spatial correlation at each specific location, ie provides a value for each location Anselin [15].
Geary’s local index ( ) is a spatial association statistic called LISA because it satisfies two criteria: The applicability for each observation to point out statistically significant spatial clusters and the ability to show that the sum of for each region analyzed is proportional to Geary global spatial association index, see Anselin [15].
Moran’s bivariate index ( ) is na index of spatial correlation between two variable (X and Y) that área obtained in spatial units, see Moran [14] and Almeida [17].
Geary’s local multivariate index as presented in Anselin [16], is defined as a multivariate statísic that is used to measure the tension between similarity of variables and similarity of geographic locations.
Lee [18] considers that the bivariate analysis seeks to odentify the existence of patterns of spatial association between two variables. In this sense the same author prospused a measuare of parametric spatial association to assess the bivariate spatial dependence.
The Gini index is defined as a measure of inequality, it is used to calculate the inequality of income distribution, see IPEA [24].
The Theil index is defined as a measure of inequality based on the uncertainty of the economic distribuition, see Theil [26].
In this work, we worked with the spatial autocorrelation of the incidence rate of the number of Covid-19 cases, using the Moran global and local index and the univariate Geary global and local index and spatial correlation of the Covid-19 incidence rate. with the following socioeconomic variables: Gini Index, Theil-L Index and the Human Development Index by Municipality. Moran's global spatial correlation index, Geary's local multivariate indices and Lee's spatial correlation index were used. In this work, spatial regression models were not used.
Point 2: The probability for a given country h to be in a class k should be the proportion of observations (households) in country h that belong to the income class k. On page 9, the first equation (it would be easier for the reader if the equation is numbered) is not exactly the proportion of people because the authors take the sum of the probability. The interpretation of the equation in not obvious. Normally, after estimating a mixture of regression model we have for each observation its estimated probabilities to be classified into the different classes identified. What is often done is to classify a given observation into the class where its estimated probability is higher. In many software this is also the method used that gives us the proportion of people in each of the classes. The authors should explain the equation on page 9 and how to interpret it. Alternatively, they may use the proportion approach which will make the interpretation easier.
Response 2: Please provide your response for Point 2. (in red)
Response 2: The equation 9 was described, defined and interpreted, the clarity of information was improved.
The Human Development Index was developed by the United Nations Development Program (UNDP), it is used to measure the human development of countries, it is measured from three dimensions: access to culture and education, standard of living and adequate income. The indicators used to measure this dimension are: literacy rate, life expectancy at birth and per capita income. Equation 9 shows the formula for the Human Development Index by Municipality which was adapted from the United Nations Development Programme, see UNDP [27].
= , (9)
em que,
: healthy life index;
: index of access to education and culture;
: adequate income standard of living index.
The Human development index by municipality is based on the estimation of economic, political, cultural and social aspects it is used to quantify the development of a certain population group, see IPARDES [22].

Reviewer 2 Report
This work is well done. I have no serious comments
Author Response
According to reviewer 2, the manuscript has finished.

Round 2
Reviewer 1 Report
The questions I had has been addressed in the revised manuscript